# The impact of the Th17:Treg axis on the IgA-Biome across the glycemic spectrum

**Heather T. Essigmann[1], Kristi L. Hoffman[2], Joseph F. Petrosino[2], Goo Jun[3], David Aguilar[3], Craig L. Hanis[3], Herbert L. DuPont[1,4], Eric L. Brown[1]\***

**1** Center for Infectious Disease, Division of Epidemiology, Human Genetics, and Environmental Sciences, University of Texas Health Science Center, Houston, TX, United States of America, **2** Alkek Center for Metagenomics and Microbiome Research, Department of Molecular Virology and Microbiology, Baylor College of Medicine, Houston, TX, United States of America, **3** Human Genetics Center, Division of Epidemiology, Human Genetics, and Environmental Sciences, University of Texas Health Science Center at Houston, Houston, TX, United States of America, **4** Kelsey Research Foundation, Houston, TX, United States of America

\* eric.l.brown@uth.tmc.edu

**Data Availability Statement:** The 16S RNA sequencing data are available in NCBI under BioProject accession number PRJNA662100. https://www.ncbi.nlm.nih.gov/bioproject/?term=PRJNA662100.

## Abstract

Secretory IgA (SIgA) is released into mucosal surfaces where its function extends beyond that of host defense to include the shaping of resident microbial communities by mediating exclusion/inclusion of respective microbes and regulating bacterial gene expression. In this capacity, SIgA acts as the fulcrum on which host immunity and the health of the microbiota are balanced. We recently completed an analysis of the gut and salivary IgA-Biomes (16S rDNA sequencing of SIgA-coated/uncoated bacteria) in Mexican-American adults that identified IgA-Biome differences across the glycemic spectrum. As Th17:Treg ratio imbalances are associated with gut microbiome dysbiosis and chronic inflammatory conditions such as type 2 diabetes, the present study extends our prior work by examining the impact of Th17:Treg ratios (pro-inflammatory:anti-inflammatory T-cell ratios) and the SIgA response (Th17:Treg-SIgA axis) in shaping microbial communities. Examining the impact of Th17:Treg ratios (determined by epigenetic qPCR lymphocyte subset quantification) on the IgA-Biome across diabetes phenotypes identified a proportional relationship between Th17:Treg ratios and alpha diversity in the stool IgA-Biome of those with dysglycemia, significant changes in community composition of the stool and salivary microbiomes across glycemic profiles, and genera preferentially abundant by T-cell inflammatory phenotype. This is the first study to associate epigenetically quantified Th17:Treg ratios with both the larger and SIgA-fractionated microbiome, assess these associations in the context of a chronic inflammatory disease, and offers a novel frame through which to evaluate mucosal microbiomes in the context of host responses and inflammation.

## Introduction

Secretory immunoglobulin A (SIgA), the most abundant immunoglobulin subtype produced by humans, maintains intestinal homeostasis by actively selecting the residents of the gut

**Funding:** National Institutes of Health grant R01DK116378 to ELB and CLH.

**Competing interests:** No authors have competing interests.

microbiota [1–3] and providing protection against pathogenic microorganisms and toxins [4, 5]. Additionally, SIgA regulates bacterial gene expression and motility, thereby impacting host metabolism, innate immune responses, and the integrity of the gut epithelium [6–11].

The fitness of the mucosal associated lymphatic tissue (MALT) (*e.g.*, the gut, bronchus, conjunctiva, lacrimal ducts, salivary ducts, nasopharynx) can be measured by the diversity of the SIgA repertoire [11, 12]. The primary purpose of the MALT is to generate and transport SIgA to these host/microbiota interfaces where they target select taxa, thereby generating distinct SIgA coated (SIgA$^+$) or SIgA uncoated (SIgA$^-$) microbial communities, *i.e.*: the IgA-Biome [12–14]. Because the quality of the SIgA response can be affected by T-cell phenotypes and mucosally-imprinted/activated plasma cells circulate across the MALT (including in the oral cavity lymphatics and Peyer's patches and lymphoid follicles of the small and large intestines) [11], we examined the impact of Th17:T-regulatory (Treg) T-cell ratios on both the salivary and gut IgA-Biomes (*i.e.*, the SIgA-coated and SIgA-uncoated microbes) in the context of diabetes phenotypes because: i) the oral cavity is the point of entry where microbes first encounter SIgA and some SIgA-coated taxa are present both in the oral cavity and the gut *e.g.*, *Lactobacillus*, *Haemophilus*, *and Prevotella* [15–17], and ii) the impact of the Th17:Treg ratios in the context of the IgA-Biome and dysglycemia has not yet been explored and will better define the impact of the Th17:Treg/SIgA axis across mucosal microbiomes (Fig 1).

Production of SIgA of high affinity and specificity is T-cell-dependent [18–22], and the nature of the T-cells populating the lamina propria and the lymphatic tissues of the gut and other mucosal sites can significantly impact the SIgA repertoire [23–26]. Without T-cells, high affinity, antigen-specific antibody responses are virtually nonexistent [27, 28], and poly-specific, cross-reactive SIgAs of low affinity and specificity dominate [19, 20, 22, 29–33]. The importance of a highly diversified (specific) SIgA repertoire in the maintenance of a complex gut microbiome is exemplified in mice with a specific mutation in activation induced cytidine deaminase (AID) that allows for antibody class switching in the absence of affinity maturation resulting in the production of normal amounts of low-affinity SIgA [34, 35]. The absence of high-affinity SIgA in such AID mutants skews the composition of the gut microbiome [36].

Immune dysregulation plays a key role in the pathogenesis and progression of type 2 diabetes [37]. Type 2 diabetes presents with impaired natural killer cell and B-cell function, abnormal proliferation of macrophages, and imbalances in the Th17:Treg T-cell axis [37–39]. Despite developing from a common precursor, Th17 and Treg T-cells have opposing inflammatory roles evident in autoimmune disease, infection, diabetes, and gut microbiome health [23, 24, 26, 40]. Tregs maintain homeostasis in part by regulating SIgA abundance and specificity, while SIgA reciprocally controls T-cell responses in the gut [41]. Skewed Th17:Treg profiles, observed in a number of chronic inflammatory conditions [40, 42], are due to both an increase in Th17 cells [23, 43–46] and a reduction in Tregs—the latter of which are vital in reversing the insulin resistance that typifies diabetes progression [46–49]. IgA production is also altered in those with hyperglycemia [21, 50, 51], with elevated serum IgA correlated with diabetes sequelae, including nephropathy, neuropathy, and vascular complications [3, 37, 50–57].

We recently conducted a preliminary study of the IgA-Biome, 16Sv4 rDNA profiling of the microbiota based on their SIgA coating, by diabetes status in a group of Mexican American adults [13]. We observed relative shifts in the SIgA$^+$ and SIgA$^-$ bacterial communities distinct from that of the overall microbiome [13], highlighting the unique perspective of IgA-Biome sequencing to identify novel relationships between the gut and salivary microbiota and the immune system in dysglycemia (those with prediabetes and diabetes). To better understand the complex dynamic between T-helper cell profiles and SIgA in the gut and saliva in the context of diabetes, we used qPCR to epigenetically quantify T-cell ratios to examine associations

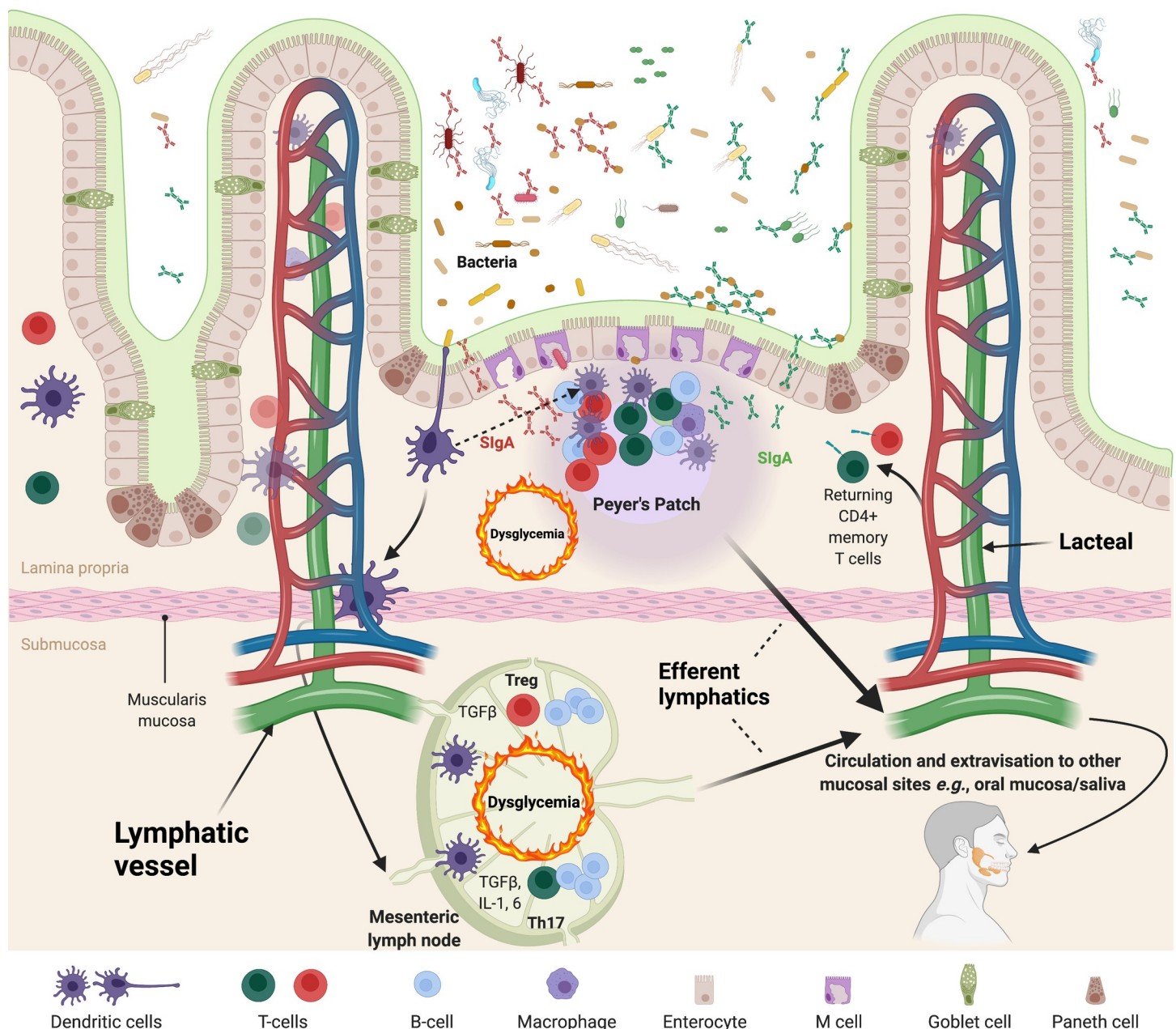

**Fig 1. The Th17:Treg SIgA axis.** IgA is produced at an approximate rate of 1 gram per day and is the primary acquired immune component secreted into mucosal surfaces [14]. Although B-cells populating these tissues can produce IgA without T-cell help (primarily in isolated lymphoid follicles; not depicted), somatic hypermutation resulting in the generation of affinity-matured (and highly specific) IgA is a product of germinal center reactions which involve T-cell help and take place almost exclusively within Peyer's patches [14]. Because T-cells play a central role in the generation of affinity matured IgAs, the type of T-cell (Th17; green cells vs. Treg; red cells) providing B-cell help can impact the quantity and quality of the mucosal IgA response (red vs. green SIgAs and their respective targets). Therefore, the Th17:Treg-SIgA axis is likely to impact the IgA repertoire secreted into mucosal surfaces that in turn shapes the architecture of mucosal microbiomes. Inflammation and associated cytokine profiles triggered by chronic disease states like diabetes (*i.e.*, dysglycemia) affect immune responses and can therefore alter Th17:Treg ratios and the resulting IgA response both in the gut and systemically. Activated T- and B-cells can migrate out of the gut via the efferent lymphatics and enter mesenteric lymph nodes or reach the circulation where they can migrate to other mucosal sites or return to the gut. T-cells in the circulation can then be isolated and Th17/Treg profiles defined. Created with Biorender.com.

between the Th17:Treg axis and the gut/salivary IgA-Biomes across the glycemic spectrum [40, 58–61]. Each qPCR assay is specific for either demethylated *FOXP3* (for Tregs) or demethylated *IL17A* (for Th17 cells) [40]. This technique provides relative and absolute immune cell

counts applicable to fresh-frozen blood samples. Signals are digital; they indicate either one positive or negative value per cell rather than arbitrarily defined thresholds for "positivity" as in flow cytometric methods limited to liquid blood. These preliminary analyses identified significant associations by taxa and between alpha and beta diversity metrics of the gut and salivary microbiomes with changes in Th17:Treg ratios in the context of both the IgA-Biome and glycemic status.

## Materials and methods

### Study population and sample collection

The Diabetes Prevention Microbiome Study is an ongoing longitudinal cohort study designed to explore the complex etiologies of type 2 diabetes and relationships with the gut, nasal, and salivary microbiomes in Mexican American adults residing in Starr County, Texas, where obesity and type 2 diabetes prevalence far exceeds U.S. averages [62]. This study was approved by the Institutional Review Board Committee for the Protection of Human Subjects of the University of Texas Health Science Center (HSC-SPH-06-0225), and informed written consent was obtained from each participant. Stool, saliva, and buffy coat samples were collected upon enrollment from an age matched subset of 24 participants, 8 with normal glycemia, 8 with prediabetes, and 8 with diabetes [13]. Characterization of the gut and salivary IgA-Biomes and demographics of the study participants and were previously published by our group [13]. Briefly, 16.67% (4/24) of subjects were male and the mean age of participants was 47.75 years (standard deviation (SD) = 5.96). The body mass index (BMI) varied by diabetes phenotype (ANOVA = 0.05) with mean BMIs of 30.6 kg/m$^2$ (SD = 4.8), 33.0 kg/m$^2$ (SD = 3.0), and 37.7 kg/m$^2$ (SD = 7.6) for those classified as normoglycemic, prediabetic, and diabetic, respectively. In the present study, the buffy coats were used to define the Th17:Treg ratio (see below) to establish correlations between the IgA-Biome and diabetes profiles.

### Diabetes classifications

Participants were defined as having normoglycemia (n = 8), prediabetes (n = 8), or diabetes (n = 8) based on the clinical cut-points for fasting glucose, 2-hour post load glucose levels, or percent glycated hemoglobin (%HbA1c) [13, 63]. Those with an fasting glucose level of 100 mg/dL– 125 mg/dL, a glucose tolerance with a 2-hour post glucose load of 140–199 mg/dL, or with %HbA1c 5.7% < HbA1c < 6.4% were classified as having prediabetes. Participants with a fasting glucose level $\geq$ 126 mg/dL, a 2-hour post load glucose level $\geq$ 200 mg/dL, or % HbA1c $\geq$ 6.5% were classified as having diabetes. Participants with blood values sufficient for classification with prediabetes or diabetes were newly diagnosed and treatment naïve to any diabetes-related medications. A definition of normoglycemic required all 3 measures be within normal range, while an abnormal value on any one test was sufficient to classify a subject as prediabetic or diabetic [13].

### IgA-Biome analyses

Bacteria from stool and saliva samples were coated with fluorescently labeled antibodies to separate IgA-coated and uncoated organisms as previously described [13]. Sorting was conducted using a BD SORP FACS Aria™ II (Special Order Research Product from Beckton Dickinson, San Jose, CA), equipped with forward scatter photomultiplier tube (FSC-PMT) for improved small particle detection, at the Baylor College of Medicine Cytometry and Cell Sorting Core [13]. Sorted samples were collected into chilled, pre-labeled epi tubes, and stored at -80˚C

until subjected to 16S rRNA gene sequencing. Most stool and saliva IgA-Biome analyses were conducted on half a million sorted bacteria although some sorts contained fewer bacteria [13].

## 16Sv4 rDNA sequencing

Genomic DNA was extracted from sorted bacterial cells using the MagAttract PowerSoil DNA kit (Qiagen) following methods adapted from the NIH Human Microbiome Project [64]. In addition to sequencing IgA+ and IgA- bacteria from each sample, unsorted (Presort) samples were collected for each sample prior to sorting and sequenced in parallel. The V4 region of the 16S rDNA gene was amplified by PCR using barcoded primers (515F, 806R) and sequenced on the MiSeq platform (Illumina) via the 2x250 bp paired-end protocol, targeting at least 10,000 reads per sample. The IgA-Biome profiles defined for the 24 participants described in the present study were previously published by our group [13].

Demultiplexed read pairs were merged using USEARCH v7.0.1090 [65] with the following parameters: merged length ≥252 bp, minimum 50 bp overlap with zero mismatches, and truncation quality >5. Merged reads were further filtered against a maximum expected error rate of 0.05 and PhiX sequences were removed. Reads were clustered into Operational Taxonomic Units (OTUs) at a similarity cutoff value of 97% using UPARSE [66] and underwent chimera removal via USEARCH [65]. OTUs were mapped against an optimized version of SILVA Database v132 [67] containing only sequences from the v4 region of the 16S rRNA gene.

## Epigenetic characterization of T-helper cell profiles

Relative counts of CD3 T-cells, Treg, Th17 subpopulations were measured by quantitative epigenetic real-time PCR at the Berlin laboratory of Precision for Medicine [58]. DNA was extracted from buffy coats at the German Cancer Research Center from frozen pellets of non-intact, nucleated blood cells (EDTA buffy coats). Buffy coats were collected at the same time stool and saliva samples for IgA-Biome analyses were collected. DNA quality was assessed using Quant-iT PicoGreen dsDNA Assay (Life Technologies) and an OD 260/280 ratio between 1.7 and 2.0 was considered acceptable. Precision for Medicine personnel were blinded to the status of the samples received. Genomic DNA was treated with ammonium bisulphite, converting unmethylated cytosine to uracil while leaving methylated cytosine unchanged, and relative quantities of different lymphocytes were determined through methylation-specific qPCR assays for unmethylated CpG sites that were shown to be stably associated with specific immune cell type lineages as described [59–61]. Absence of CpG methylation in those gene loci (amplicon regions) was used for the epigenetic cell counting. These loci were *CD3G/ CD3D*, *FOXP3*, and *IL-17A* for quantification of total CD3+ T-cells, regulatory T-cells (Tregs), and Th17 [59–61]. Total cell count was determined by methylation-specific qPCR for the *GAPDH* gene locus as described in Sehouli *et al.* [60]. The relative content of CD3 T-cells, Treg, and Th17 cells in the sample was determined by calculating the ratio of the respective three loci and the number of unmethylated *GAPDH* genes in the sample. In addition, the relative content of Treg and Th17 cells within the overall CD3 T-cell compartment was determined by calculating the ratios of unmethylated copies for *FOXP3/CD3* and *IL-17A/CD3* [58, 60, 61]. In addition, the relative composition of pro-inflammatory Th17 cells and immune suppressive Treg cells was calculated based on the copy numbers of unmethylated *IL-17A/FOXP3* [58, 60, 61]. All measurements were run in duplicate and subjected to rigorous quality controls according to the laboratory's quality management system that is accredited under ISO 17025. All plate runs conformed to performance limits set for standards, calibrators, and references that monitor standard linearity, bisulfite conversion, and qPCR efficiency. No samples failed

quality control and the buffy coat samples used for the present study fell within established standards set by a healthy control cohort.

## Statistical analysis

A two tailed t-test and ANOVA were used to assess Th17:Treg ratios by diabetes phenotypes using Stata 16 statistical software (Statacorp, College Station, TX). As described previously [13], the Agile Toolkit for Incisive Microbial Analyses 2 (ATIMA2) was used as an integrated solution for analyzing and visualizing microbiome data (https://atima.research.bcm.edu). Analyses of alpha diversity metrics were performed in ATIMA2 (supplementary figures) and in Stata 16 using ordinary least squares regression adjusted for body mass index and graphed using GraphPad Prism version 8.0 (San Diego, CA). Comparisons of community dispersion were assessed using unweighted UniFrac distance and visualized using ATIMA2 and statistical significance determined via Mann-Whitney U test or PERMANOVA adjusted for body mass index using the adonis and betadisper functions in the R package vegan [68, 69]. Differentially abundant taxa by Th17:Treg ratios were determined via Linear discriminant analysis effect size (LEfSe) [70] using the Galaxy web platform with parameters of alpha of 0.05 and threshold of logarithmic linear discriminative analysis (LDA) score of 2.0 [71]. Analyses using LEfSe were limited to taxa present in at least 10% of samples and visualized using GraphPad Prism version 8.0.

## Results

### Epigenetic quantification of the Th17:Treg ratio

Relative Th17 and Treg cell counts were defined for individuals with previously established stool and salivary IgA-Biomes using quantitative epigenetic real-time PCR and the Th17:Treg ratios calculated (Fig 2) [13]. Comparisons of Th17:Treg ratio between individuals with normoglycemia (n = 8), prediabetes (n = 8), or diabetes (n = 8) identified similarly elevated Th17:Treg ratios in dysglycemic individuals, those classified as having prediabetes (mean ± standard deviation (SD): 2.22±0.20; range: 1.514–3.239) and diabetes (mean ± SD: 1.96±0.19; range: 1.428–3.222), compared to individuals with normal blood sugar values (*i.e.*, normoglycemia; mean ± SD: 1.58±0.16; range: 0.8177–2.006). For this reason, those with prediabetes and diabetes were combined into a dysglycemic category for some of the analyses. The Th17:Treg ratios of those with dysglycemia were significantly higher than those without dysglycemia (Fig 2A; p<0.042). Analyses of differences across the three diabetic phenotypes approached the margin of statistical significance (Fig 1B; ANOVA, p<0.068).

Th17:Treg T-cell ratios were normally distributed (Shapiro-Wilk test of normality p-value = 0.08) with a median of 1.79 and mean of 1.90 (SD: 0.57, range: 0.82–3.24). Consequently, and because there is no established definitions for what constitutes high or low Th17:Treg ratios, the 24 participants were dichotomized into 2 categories above and below the median Th17:Treg ratio, indicative of a (pro-inflammatory profile or anti-inflammatory profile, respectively, for analyses requiring categorical characterization (beta diversity, LEfSe) [40, 72].

### Th17:Treg ratios and glycemic status affect alpha and beta diversity

The impact of Th17:Treg ratios on the microbiota was first assessed independently of the IgA--Biome by analyzing the Presort samples (*i.e.* stool and saliva samples not subjected to SIgA$^+$ or SIgA$^-$ sorting) across the diabetes spectrum. Analyses of alpha diversity, an evaluation of within sample diversity, did not identify differences in bacterial evenness (Shannon diversity

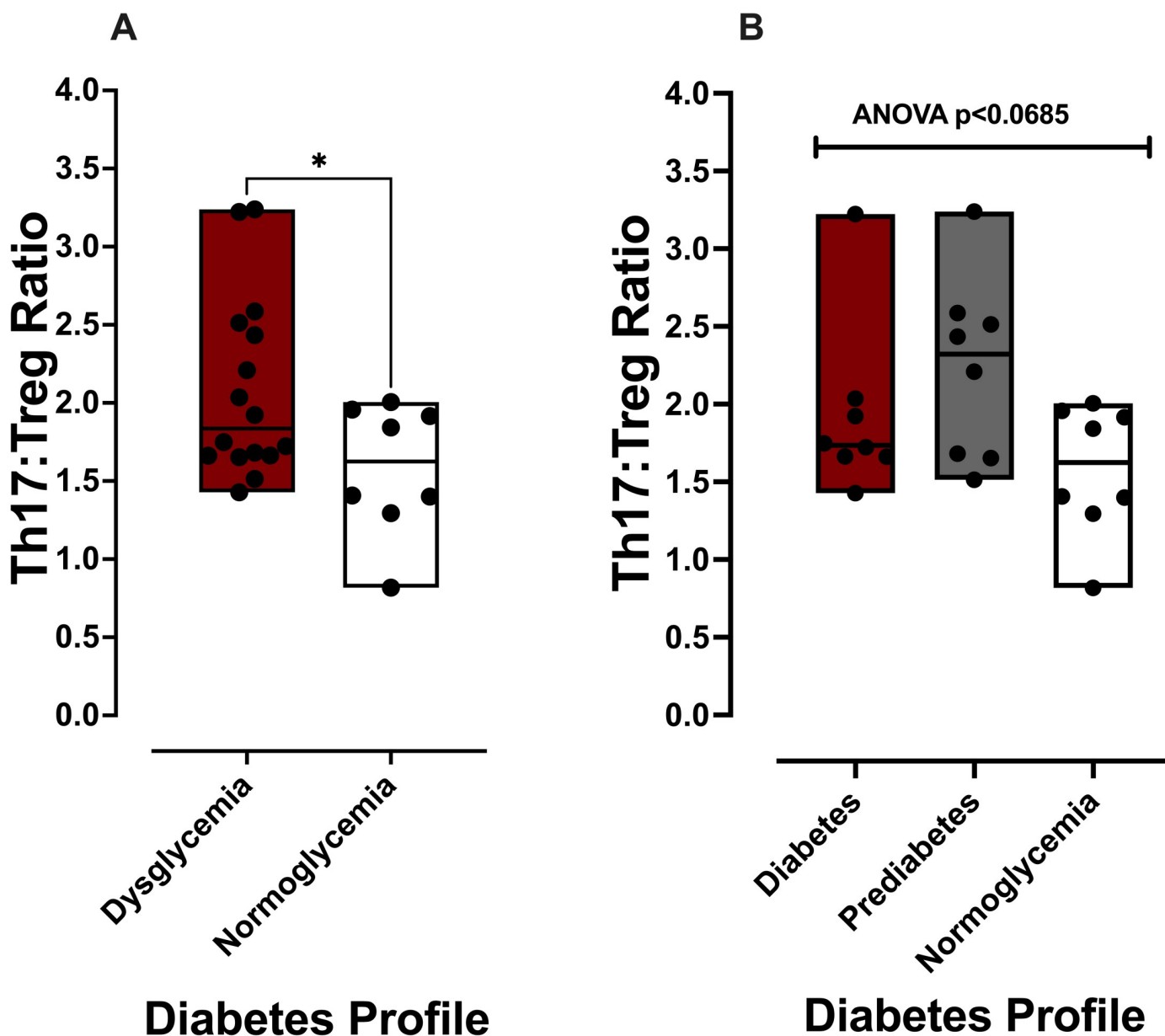

**Fig 2. Th17:Treg ratios by diabetes profile.** Box plots depict the median (center line in each box) and box boundaries represent the range. (**A**) Th17:Treg ratio comparisons between those with normoglycemia and dysglycemia; *p<0.042 two-tailed t-test. (**B**) Th17:Treg ratio comparisons between those with normoglycemia, prediabetes, and diabetes; ANOVA p<0.0685.

index) between those with normoglycemia or dysglycemia for either the stool or salivary microbiomes by T-cell ratio. Bacterial richness (observed OTUs [operational taxonomic units]) of stool, however, significantly increased with increasing Th17:Treg ratios in those with dysglycemia (Fig 3A; p<0.005). Although not statistically significant, the opposite trend, *i.e.*, decreasing richness with increasing Th17:Treg ratio, was observed in those with normoglycemia (Fig 3A; p<0.066). When the same analyses were conducted across the three diabetes phenotypes those with prediabetes and diabetes followed a similar trend *i.e.*, increasing OTUs with increasing Th17:Treg ratios (p<0.063 and p<0.035, respectively) (Fig 3B). Analyses of salivary samples did not identify statistically significant differences across the glycemic

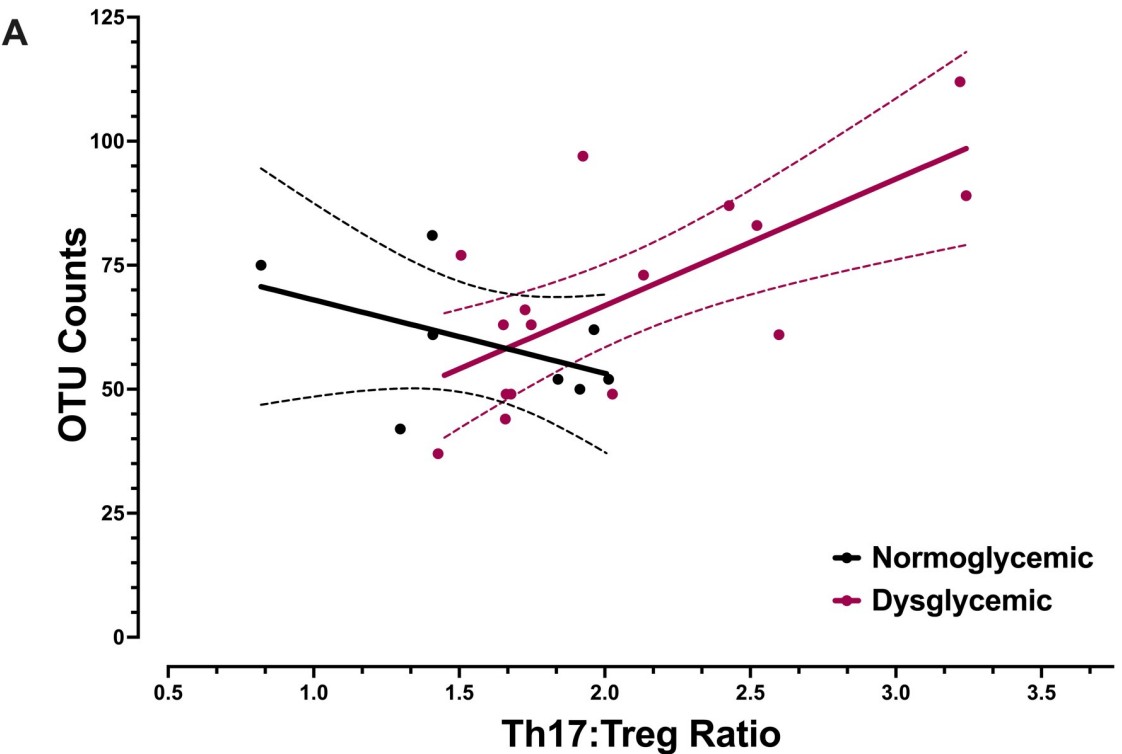

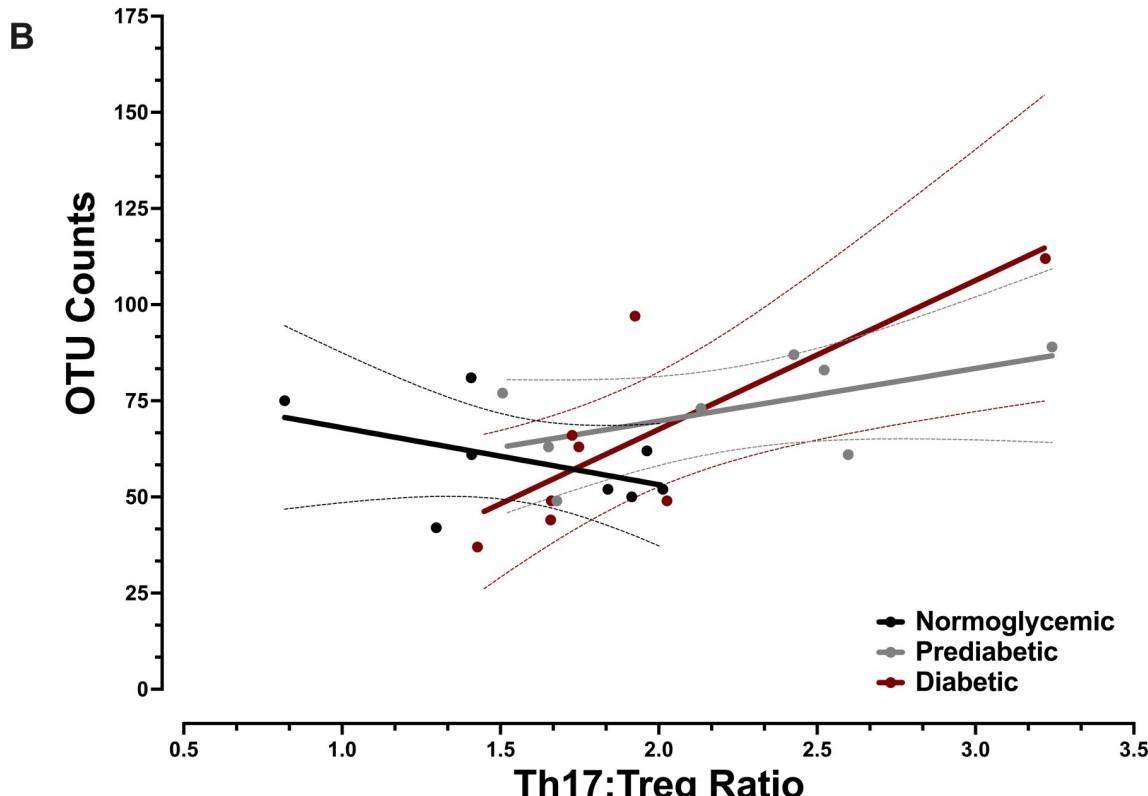

**Fig 3. Impact of Th17:Treg ratios on alpha diversity of the stool microbiome by diabetes phenotype.** Linear regression of OTU counts from Presort stool samples on Th17:Treg ratios by (A) normoglycemia and dysglycemia profiles, p-value of <0.066

(normoglycemic) and <0.005 (dysglycemic) or (B) by normoglycemia, prediabetes, and diabetes profiles, p-value of p<0.063 (prediabetes) p-value of p<0.035 (diabetes). P-values obtained using ordinary least squares regression adjusted for body mass index and result from a two-sided hypothesis.

spectrum; however, a similar trend to that observed with stool samples was present, *i.e.*, directly and inversely proportional relationships between OTU counts and Th17:Treg ratios for those with dysglycemia and normoglycemia, respectively (S1A and S1B Fig).

Comparisons of individuals grouped as having pro-inflammatory or anti-inflammatory profiles based on Th17:Treg ratios were used to investigate beta, or between sample, diversity of stool and salivary Presort samples using unweighted UniFrac distances, a distance metric that incorporates phylogenetic relatedness and multivariate statistical techniques to determine whether microbial communities are significantly different. Principle coordinate analysis of unweighted Unifrac distances identified significant differences in community composition between individuals with different inflammatory phenotypes in both stool and saliva samples of those with dysglycemia (Fig 4B and 4D; p<0.021 and p<0.036, respectively). Significant differences in the stool or saliva Presort samples of those with normoglycemia were not observed (Fig 4A and 4C). Analyses by prediabetes and diabetes phenotypes identified significant differences in the stool microbiome of those with prediabetes (S2 Fig; p<0.03). No significant differences in the salivary microbiome in those with prediabetes or diabetes were observed (S3 Fig).

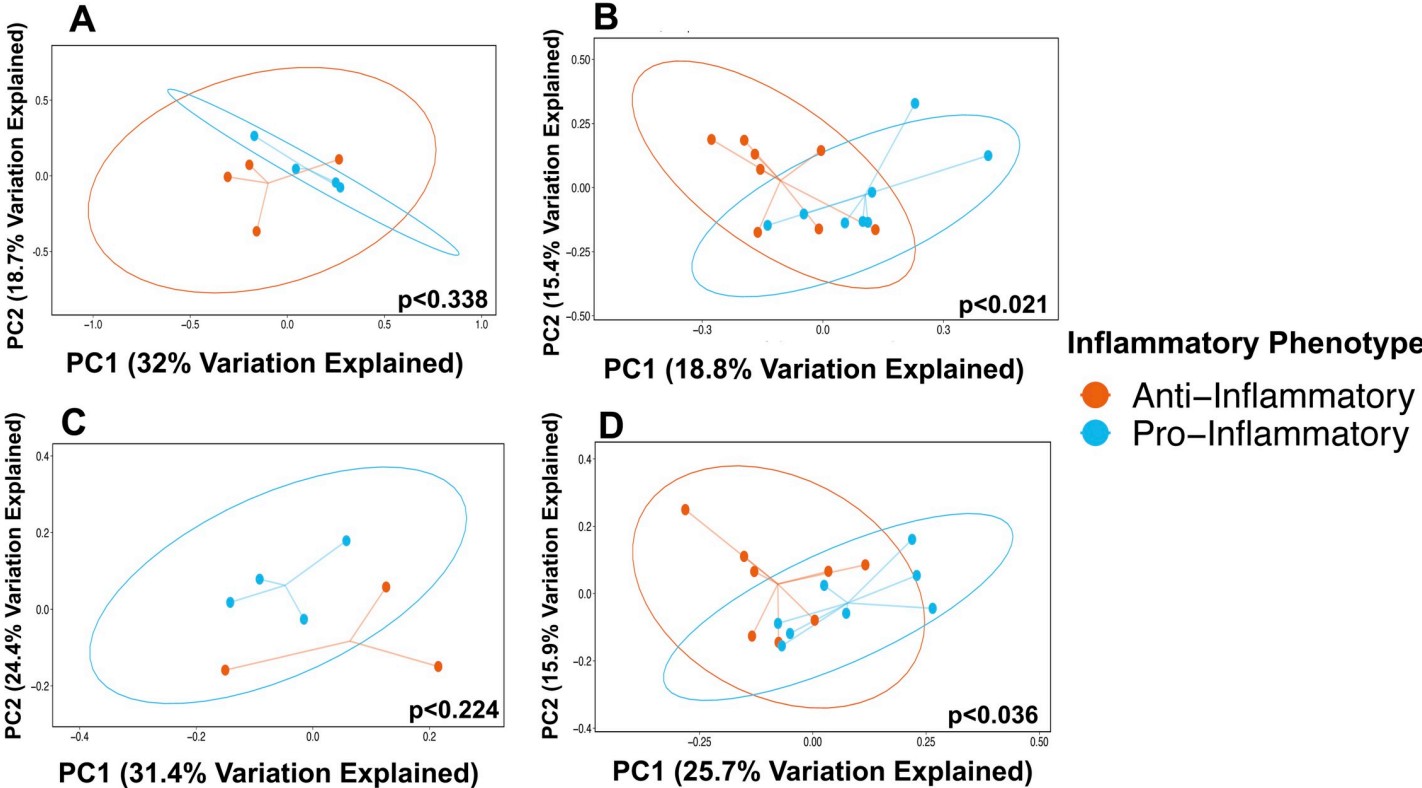

**Fig 4. Principle coordinate analyses of UniFrac distances of stool and saliva Presort samples by Th17:Treg ratios in those with normoglycemia and dysglycemia.** Principal coordinate analyses of unweighted UniFrac distances of stool (**A**, **B**) and (**C**, **D**) saliva Presort samples in those with (**A**, **C**) normoglycemia and (**B**, **D**) dysglycemia. P-values determined using PERMANOVA and include adjustments for body mass index.

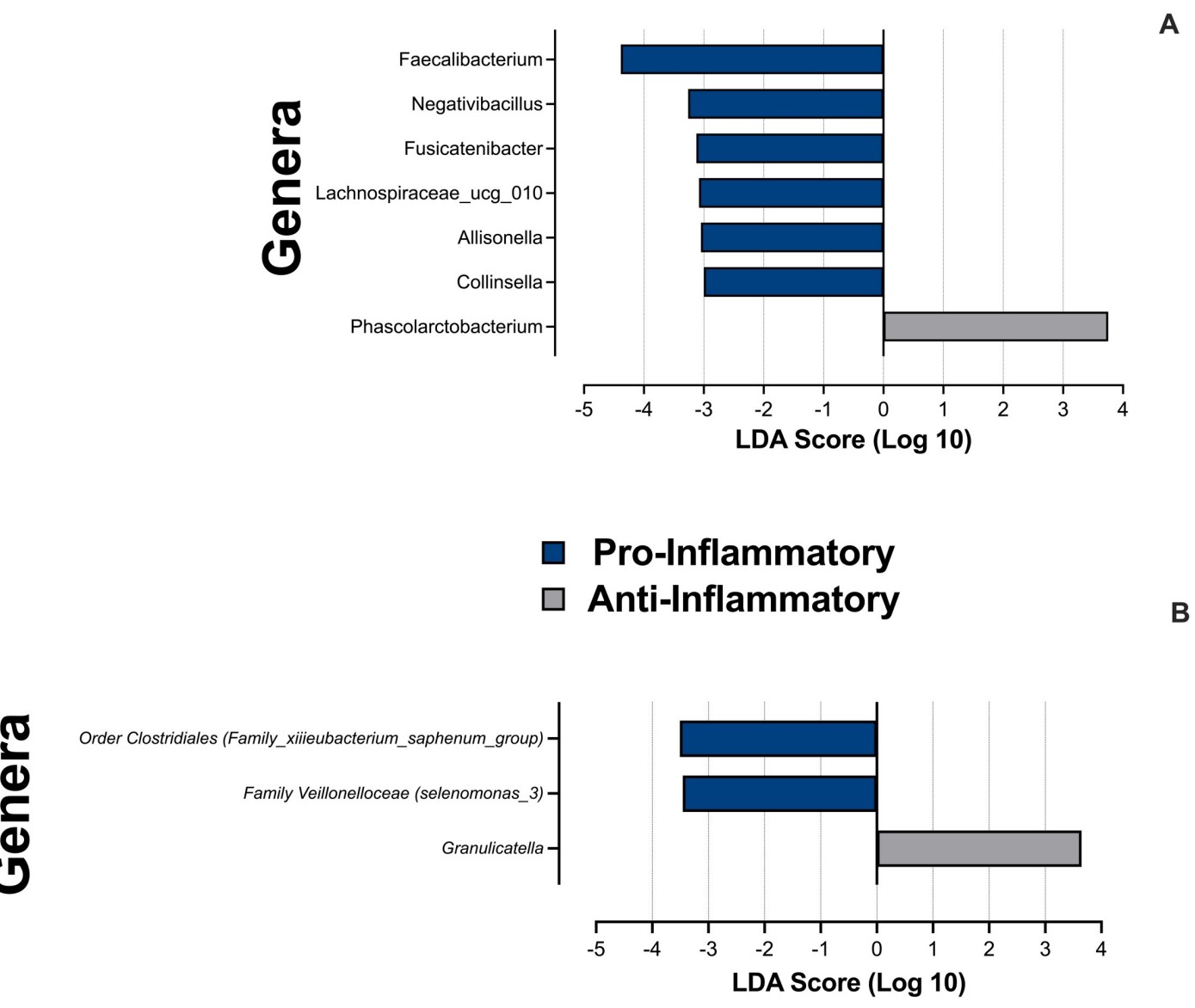

**Fig 5. LEfSe of stool and salivary Presort samples by inflammatory profiles.** LEfSe identified genera (or lowest taxonomic designation available) significantly associated with pro- or anti-inflammatory Th17:Treg profiles in (**A**) stool and (**B**) saliva Presort samples. Analyses were conducted using parameters α<0.05 and linear discriminant analysis (LDA) threshold ≥2.0.

We used the biomarker discovery tool Linear Discriminant Analysis Effect Size (LEfSe) [73] to identify preferentially abundant genera in the stool and saliva Presort samples of individuals classified as having pro- or anti-inflammatory profiles (Fig 5). Six organisms were found to be differentially abundant of the stool of those with pro-inflammatory Th17:Treg ratios, including members of the genus *Faecalibacterium* (p = 0.008), while only one genus, *Fusicatenbacter* (p = 0.039), was found to be differentially abundant of the stool of those with an anti-inflammatory profile. In saliva, *Granulicatella* (p = 0.031) and members of genera *Selenomonas* (p = 0.036) and *Eubacterium* (p = 0.040) were preferentially abundant in the Presort salivary microbiome of those with anti- and pro-inflammatory Th17:Treg profiles, respectively.

## Associations between the IgA-Biome, Th17:Treg ratios, and glycemic status

The observation that almost all human SIgA protein sequences show evidence of somatic hypermutation and affinity maturation indicates that T-cell-dependent responses likely play a dominant role in shaping the IgA-Biome. More specifically as it relates to IgA production, B-cells of the mucosal lymphoid tissues are different form B-cells populating non-mucosal lymphoid tissues in that they can undergo class switching from IgM/IgD to IgA without T-cell help; however, like B-cells at all other anatomic sites, mucosal B-cells cannot produce high affinity antibodies, the products of germinal center reactions and somatic hypermutation, without the aid of T-cells [11, 74]. We therefore evaluated the interaction between Th17:Treg ratios and the IgA-Biome in the context of glycemic status. Linear regression of alpha diversity metrics on Th17:Treg ratios by stool IgA-Biome compartments (SIgA$^+$ and SIgA$^-$) identified significant increases in richness (OTU count) with increasing Th17:Treg ratios in those with dysglycemia in both compartments (Fig 6A; p<0.001). When the analysis of Th17:Treg ratios and the IgA-Biome was conducted across the three diabetes phenotypes similar trends were observed (Fig 6B; p<0.132 and p<0.018 in the IgA$^+$ and IgA$^-$ compartments, respectively of those with prediabetes and p<0.012 and p<0.022 in the IgA$^+$ and IgA$^-$ compartments, respectively of those with diabetes). The opposite trend, although not significant, was observed in normoglycemic individuals, whereby richness decreased in both SIgA compartments as the Th17:Treg ratio increased (Fig 6). Analysis of alpha diversity metrics in the salivary IgA-Biome did not identify significant associations (S4A and S4B Fig).

Principle coordinate analyses of unweighted UniFrac distances (Fig 7) identified significant differences in microbial community composition by Th17:Treg ratio-derived inflammatory phenotypes in the SIgA$^+$ stool samples of those with dysglycemia (Fig 7B; p<0.003). No differences by inflammatory phenotype were observed in those without diabetes in the SIgA$^-$ fraction of stool samples (Fig 7A) nor in either SIgA compartment of the salivary microbiome. Within the dysglycemic group, similar analyses comparing inflammatory phenotype of those with prediabetes and diabetes identified significant differences only in the stool SIgA$^+$ and SIgA$^-$ compartments of those with prediabetes (S5 Fig). No significant differences in the salivary IgA-Biome by inflammatory phenotype were observed in subjects with prediabetes or diabetes (S6 Fig).

Comparisons across diabetes phenotypes of unweighted UniFrac distances between each subject within an inflammatory profile, however, revealed significant differences in community composition within the SIgA compartments of stool (Fig 8) and saliva (S7 Fig). The gut microbial communities from individuals with a pro-inflammatory phenotype within the SIgA$^+$ fraction of those with normoglycemia (p<0.0152) and the SIgA$^-$ fraction in those with dysglycemia (Fig 8; p<0.023) were significantly less diverse than those with a lower inflammatory profile. In saliva, independent of SIgA coating (*i.e.*, Presort sample) the microbial communities of normoglycemic individuals with a pro-inflammatory profile were more diverse than those with an anti-inflammatory profile (S7 Fig).

LEfSe also identified preferentially abundant genera between individuals with Th17:Treg ratios above and below the median value in both the stool and salivary IgA-Biomes (Fig 9). Similar to what was observed for Presort samples, the genus *Faecalibacterium* had the highest LDA score in subjects with a pro-inflammatory phenotype in the SIgA$^+$ compartment (p = 0.014). The similarity between taxa identified in both the Presort (Fig 5) and SIgA$^+$ fractions (Fig 9), compared to the taxa present in the SIgA$^-$ fraction, suggested SIgA$^+$ taxa drive the significant associations observed in the Presort analyses. Only *Allinsonella* was present in both the Presort (p = 0.018) and SIgA$^-$ (0.014) fractions and *Faecalibacterium* was present across all three compartments (Fig 9). Dominant taxa in the salivary IgA-Biome included

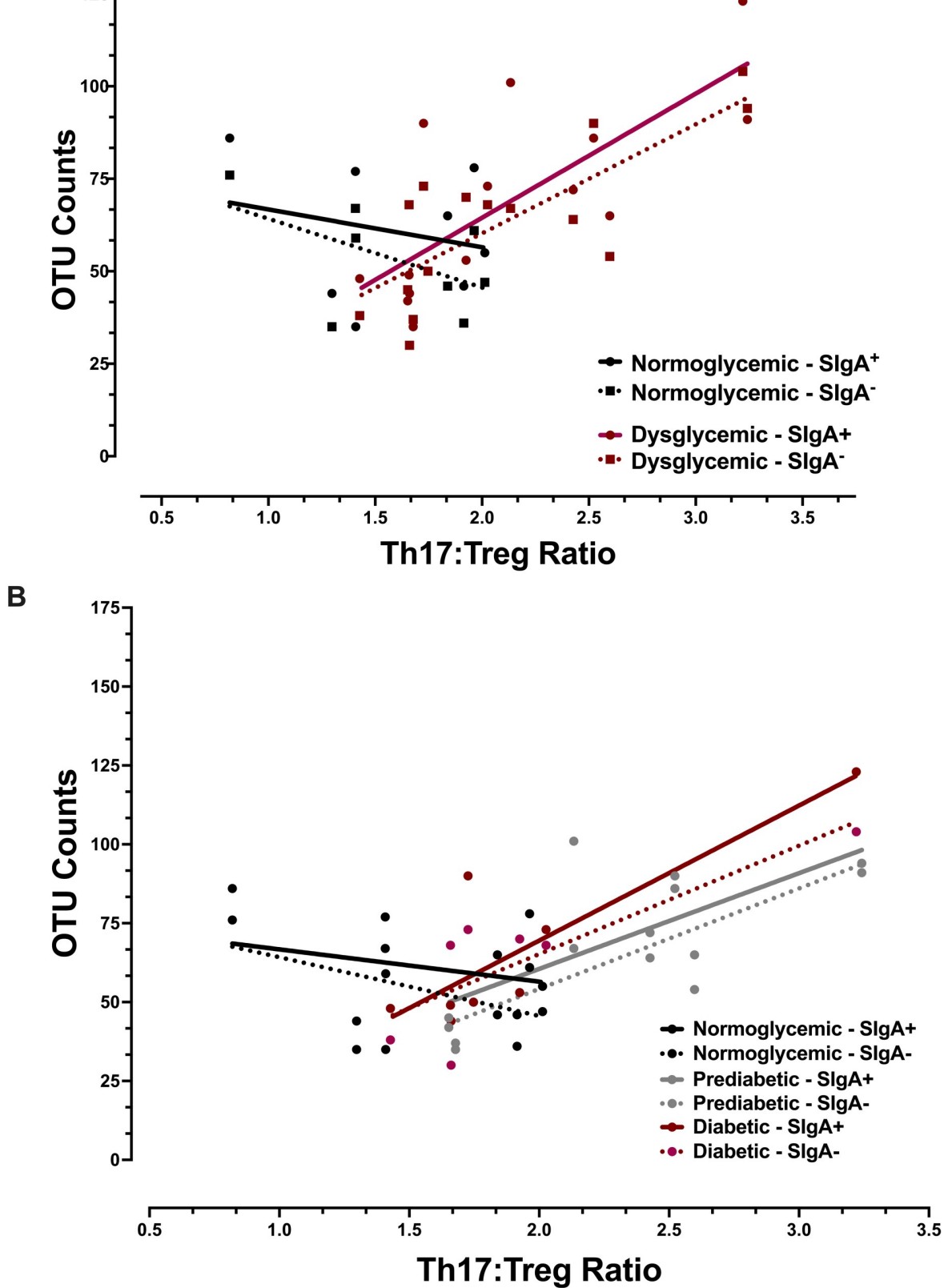

**Fig 6. Impact of Th17:Treg ratios on alpha diversity of the stool IgA-Biome by diabetes phenotype.** (**A**) Linear regression of OTU counts on Th17:Treg ratios in individuals with normoglycemia and dysglycemia. p<0.314 and p<0.118 in the IgA$^+$ and IgA$^-$ compartments, respectively, for those in the normoglycemic group and p<0.001 for both the IgA$^+$ and IgA$^-$ compartments for those in the dysglycemic group. (**B**) Linear regression of OTU counts on Th17:Treg ratios in those individuals with normoglycemia, prediabetes, and diabetes. p<0.132 and p<0.018 in the IgA$^+$ and IgA$^-$ compartments, respectively, for individuals with prediabetes and p<0.012 and p<0.022 in the IgA$^+$ and IgA$^-$ compartments, respectively, for individuals with diabetes. P-values obtained using ordinary least squares regression adjusted for body mass index and result from a two-sided hypothesis.

*Mogibacterium* in the SIgA$^+$ (p = 0.038) compartment of those with a pro-inflammatory phenotype. No significant taxa in the SIgA$^-$ compartment in those with a pro-inflammatory phenotype were identified. Analysis of individuals classified as having an anti-inflammatory phenotype identified *Cardiobacterium* (p = 0.032) and *Selenomonas* (p = 0.041) as preferentially abundant in the SIgA$^+$ compartment and *Gemella* (p = 0.007), *Oribacterium* (p = 0.034), and *Stomatobaculum* (p = 0.024) in the SIgA$^-$ compartment.

## Discussion

The relationship between the gut microbiome and human health has become increasingly evident [2, 11, 25, 37, 75–82]. We now recognize that disturbances of the gut microbiota trigger ripple effects that impact immunoglobulin production, T-helper cell responses, vaccine efficacy, infection [2, 8, 21, 83–88], metabolism [89–91], nutrition [92], and even the composition of microbiota in the lungs and oral mucosa [11, 21, 93–96]. That the gut microbiome can influence inflammation and host metabolism in a manner consistent with what is observed at the onset of type 2 diabetes, intimates a possible causal role of the gut microbiome in type 2 diabetes initiation or progression [97, 98]. Therefore, identifying the microbial constituents that have the host's immunological attention or, equally important, organisms that should have the host's attention but for unknown reasons do not, is critical to fully understanding the complex etiology of type 2 diabetes.

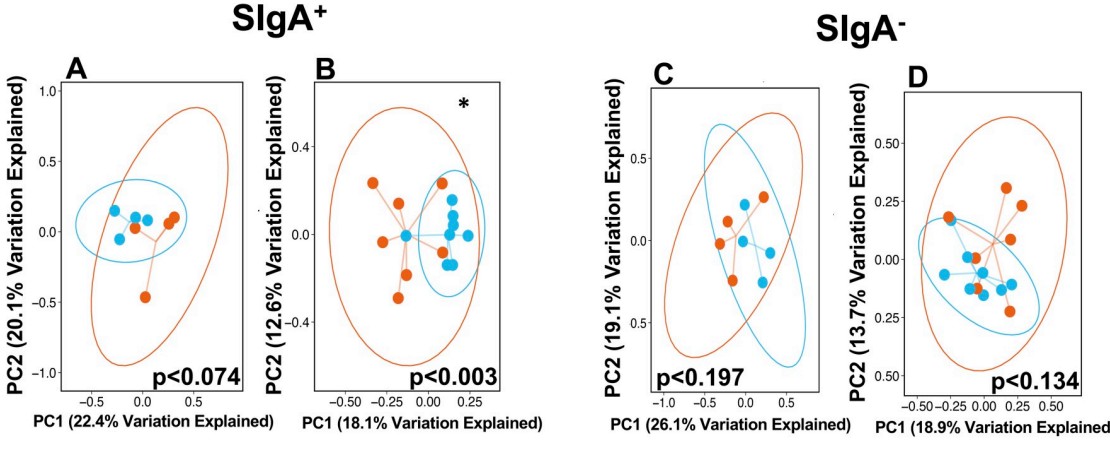

**Fig 7. Principal coordinate analyses of UniFrac distances of the stool IgA-Biome by Th17:Treg ratio-derived inflammatory phenotypes across diabetes status.** Principal coordinate analyses of unweighted UniFrac distances of the (**A, B**) SIgA$^+$ and (**C, D**) SIgA$^-$ compartments of individuals with differing inflammatory profiles. (**A, C**) Normoglycemic and (**B, D**) dysglycemic. Statistical significance was determined using PERMANOVA adjusted for body mass index; *p<0.003.

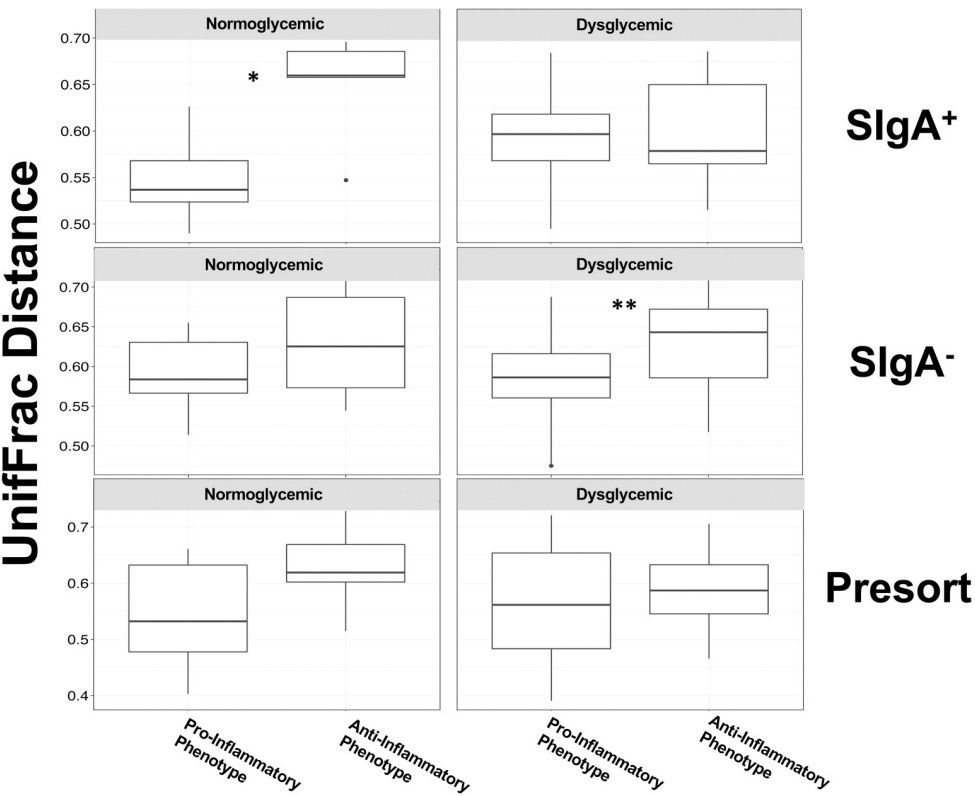

**Fig 8. Comparisons of UniFrac distances between of the stool IgA-Biome of normal and dysglycemic individuals categorized by inflammatory phenotype.** Box plots compare unweighted UniFrac distances between all individuals within each inflammatory phenotype, respectively, by SIgA compartment and glycemic profile. Normoglycemic subjects with a pro-inflammatory phenotype have SIgA$^+$ bacterial community compositions more similar to each other than do those with an anti-inflammatory phenotype ($^*$p<0.0152). Similarly, dysglycemic subjects with a pro-inflammatory phenotype have SIgA$^-$ bacterial community compositions more similar to each other than do those with an anti-inflammatory phenotype ($^{**}$p<0.023). P-values determined using the Mann-Whitney U-test.

In the present study we aimed to identify associations across the Th17:Treg-SIgA axis and glycemic profiles by studying a population whose stool and salivary IgA-Biomes had been previously characterized [13]. Even with the small sample size examined, these preliminary analyses identified significant differences in alpha and beta diversity metrics and taxonomy across T-cell ratios. Differences were most notable in the stool IgA-Biome using both continuously measured and dichotomized Th17:Treg ratios, with categorical classification based on the premise that higher ratios corresponded to higher inflammatory profiles [40, 72].

Analysis of within-subject variation revealed significant increases in bacterial richness (OTU counts) with increasing Th17:Treg ratios in those with dysglycemia (*e.g.*, prediabetes and diabetes phenotypes combined). When the dysglycemic group was separated into those with prediabetes and diabetes, the association between Th17:Treg ratios and OTU counts persisted. These significant effects held across the unsorted Presort sample and both SIgA compartments (SIgA$^+$ and SIgA$^-$) demonstrating that bacterial richness positively correlated with measures of inflammation only in those with abnormal blood sugar values. Although analyses of saliva did not identify significant associations between relative abundance and Th17:Treg ratios, it should be noted that same trend for those with dysglycemia (increasing Th17:Treg rations correlated with increasing OTUs) and normoglycemia (increasing Th17:Treg rations

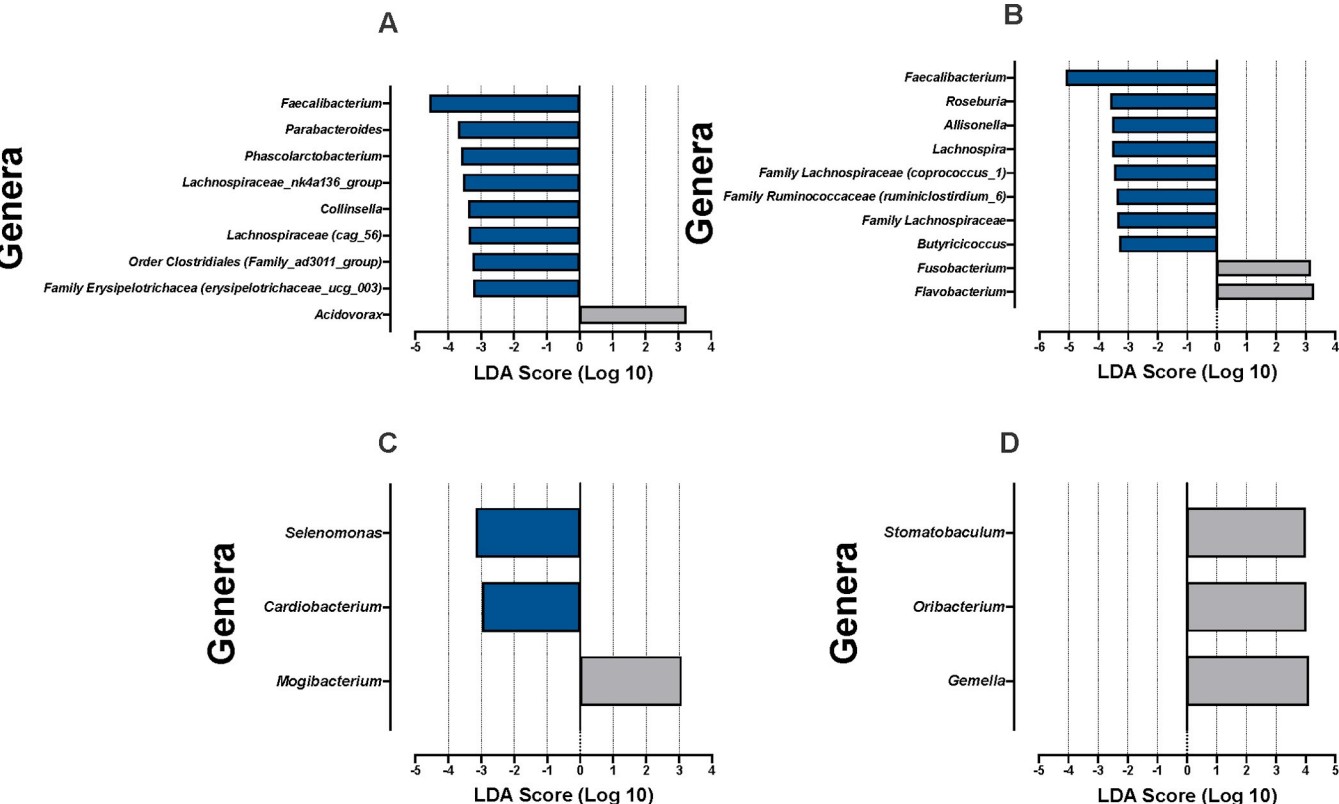

**Fig 9. LEfSe of stool and salivary IgA-Biomes by Th17:Treg ratio-derived inflammatory phenotypes.** LEfSe identified genera (or next taxonomic designation available) associated with the Th17:Treg ratios in (**A**) stool SIgA+, (**B**) stool SIgA-, (**C**) saliva SIgA+, or (**D**) saliva SIgA- compartments. Analyses were conducted using parameters α<0.05 and linear discriminant analysis (LDA) threshold ≥2.0.

correlated with decreasing OTUs) was observed. This pattern implicates a roll for Th17 and Treg cells in shaping the microbiota across the MALT.

The premise that increasing diversity is associated with health may therefore need re-evaluation, particularly in the context of the IgA-Biome [99, 100]. That is, increased diversity may not be beneficial, particularly in SIgA+ or SIgA- fractions, if SIgA responses are focused on microbiota critical to gut homeostasis or if the increase in diversity is the result of an outgrowth of pathobionts associated with disease as observed in some irritable bowel disease patients [11]. A recent study that observed healthy ageing and longevity was associated with reduced microbial diversity and loss of specific taxa further disrupts the paradigm that increased diversity is always healthful [101].

We also observed significant differences in community composition (beta diversity) between individuals categorized by their inflammatory profiles in the stool and salivary microbiomes of those with dysglycemia (Fig 4). In stool, these changes were most strongly associated with the SIgA+ compartment (Fig 7) and may implicate SIgA-coated organisms in the pathological processes associated with higher Th17:Treg ratios and inflammation [102]. A similar but not significant trend in the SIgA+ fraction of those with normal blood sugar values may indicate that changes to SIgA-targeted communities is heightened in the context of an inflammatory disease like type 2 diabetes. It is also remarkable that even in this small data set, a general trend was observed, particularly in stool IgA-Biome compartments and the Presort samples towards decreasing community diversity (*i.e.*, lower UniFrac distance) in individuals with a more inflammatory T-cell profile independent of glycemic status (Fig 8). This suggested

that individuals with a higher inflammatory profile have communities more similar to each other compared to those with a lower inflammatory profile (Fig 8). This general trend was not observed for the salivary IgA-Biome. In fact, the Presort of those with normoglycemia had the opposite association *i.e.*, less diversity between salivary communities of those with reduced inflammatory profiles compared to those with higher inflammatory profiles (S7 Fig).

Preferentially abundant genera across the IgA-Biome in the context of inflammatory phenotypes based Th17:Treg ratios were identified using LEfSe. In the stool Presort samples, genera significantly associated with pro-inflammatory profiles appeared to be driven by SIgA-targeted organisms since many taxa (*Colinsella*, *Lachnospiraceeae*, *Phascolarcitobacterium*, and *Faecalibacterium*) were found enriched in both the Presort and SIgA$^+$ fractions. *Parabacteroides* and *Erysipelotrichaceae* were unique to the SIgA$^+$ fraction and *Allinsonella* and *Negativibacillus* unique to the stool Presort. The only genus present in both SIgA$^+$ and SIgA$^-$ compartments was *Faecalibacterium*, a genus that only contains 22 species, 12 of which inhabit the human gut [103]. Although *F. prausnitzii* has been associated with chronic inflammation, other *Faecalibacterium* species are leading candidates for the development of probiotics and biotherapeutics [103, 104]. It is possible that different *Faecalibacterium* species/strains with varying levels of antigenicity may be responsible for the presence of this species in both the SIgA$^+$ and SIgA$^-$ compartments; similar to the two strains of *Bacteroides fragilis* described by Palm *et al.*, one associated with irritable bowel disease and the other with a healthy gut [102]. Identification of *Collinsella* was also interesting since increasing relative abundance of this organism has previously been positively correlated to diabetes and negatively correlated to weight loss, suggesting that manipulation/reduction of diabetes-associated taxa such as *Collinsella* may be an avenue through which to improve glucose homeostasis [105–107].

Although the predominance of our findings involved the gut, developing an understanding of the salivary IgA-Biome in the context of health and disease may be as important since *i*) the oral mucosa is the first point of contact between SIgA and the microbiota, *ii*) SIgA responses stimulated at distinct mucosal sites recirculate between compartments thereby affecting each site's SIgA responses, *iii*) many SIgA-targeted oral microbiota can be detected in the gut, and *iv*) oral dysbiosis impacts gut microbiome homeostasis [11, 108]. Although the differences observed in the salivary IgA-Biome in the present analyses were not as striking as those described for the gut IgA-Biome, our small sample size and the fewer number of taxa that colonize the oral mucosa (relative to the number colonizing the gut) may have muted our sensitivity to alterations in salivary microbiota with changing Th17:Treg ratios [108]. Nevertheless, we observed significant changes in salivary IgA-Biome beta diversity between individuals dichotomized by Th17:Treg ratio in those with dysglycemia. This supported the hypothesis that inflammatory changes as a consequence of changing Th17:Treg ratios and disease status have fundamental impacts across mucosal-associated lymphoid tissues and that interventions targeting multiple mucosal sites as a means of re-establishing systemic symbiosis across mucosal compartments may be possible.

The role of the gut microbiota in shaping and maintaining immune health cannot be overstated. Gut microbial communities significantly impact the Th17:Treg balance, an association easily appreciated in germ free mice that present with impaired Treg function [24, 100]. Conversely, in both mice and humans, Treg deficiency resulted in chronic inflammation and the elicitation of autoimmune responses in various organs [100] and specific strains of *Bacteroides fragilis* and *Clostridia* species have been observed as key drivers of Treg development and suppressors of Th17 responses [24]. We propose a scenario where host factors (*e.g.*, genetics, SIgA) and the environment (*e.g.*, diet, antibiotics, infections) shape a core microbiota. The microbiota, via interactions with immune components such as innate immune receptors in the gut and dendritic cells (in addition to other antigen presenting cells of the lamina propria), establish a cytokine milieu that shapes the T-helper cell profiles of the mucosa. B-cells

responding to the same microbial stimuli will in turn receive help signals from activated T-cells required for the establishment diverse and specific SIgA. The cycle is completed when SIgA interacts with and re-shapes the microbiota that originally stimulated its generation [11].

Our preliminary results demonstrated that an association exists between dysglycemic status and epigenetically quantified Th17:Treg ratios across the SIgA Biomes. This pilot project is the first of its kind to associate epigenetically quantified Th17:Treg ratios with both the larger and SIgA-fractionated microbiome, assess these associations in the context of a chronic inflammatory disease with increasing global prevalence and offers a novel frame through which to evaluate mucosal microbiomes in the context of host responses and inflammation. Larger sample sizes and deeper analyses into the specific species/strains associated with these phenotypic changes are needed to more fully illuminate the effects of inflammation on the microbiome and will be crucial to developing interventions designed to alter microbial populations as a means of establishing gut/immune homeostasis that in turn drive systemic health.

## Supporting information

**S1 Fig. Impact of Th17:Treg ratios on alpha diversity of the salivary microbiome by diabetes phenotype.** Linear regression of OTU counts from Presort stool samples on Th17:Treg ratios by (**A**) normoglycemia and dysglycemia profiles, p-value of <0.066 (normoglycemic) and <0.198 (dysglycemic) or (**B**) by normoglycemia, prediabetes, and diabetes profiles, p-value of p<0.136 (prediabetes) p-value of p<0.381 (diabetes). P-values obtained using ordinary least squares regression and result from a two-sided hypothesis.
(TIFF)

**S2 Fig. Principle coordinate analyses of UniFrac distances of stool Presort samples by Th17:Treg ratios in those with normoglycemia, prediabetes, and diabetes.** Principal coordinate analyses of unweighted UniFrac distances of the stool microbiomes of those with (**A**) normoglycemia, (**B**) prediabetes, or (**C**) diabetes. P-values determined using PERMANOVA.
(TIFF)

**S3 Fig. Principle coordinate analyses of UniFrac distances of saliva Presort samples by Th17:Treg ratios in those with normoglycemia, prediabetes, and diabetes.** Principal coordinate analyses of unweighted UniFrac distances of the stool microbiomes of those with (**A**) normoglycemia, (**B**) prediabetes, or (**C**) diabetes. P-values determined using PERMANOVA.
(TIFF)

**S4 Fig. Impact of Th17:Treg ratios on alpha diversity of the saliva IgA-Biome by diabetes phenotype.** (**A**) Linear regression of OTU counts on Th17:Treg ratios in those individuals with normoglycemia and dysglycemia. Normoglycemic p<0.958 and p<0.359 in the IgA$^+$ and IgA$^-$ compartments, respectively, and dysglycemic p<0.715 and p<0.533 in the IgA$^+$ and IgA$^-$ compartments, respectively. (**B**) Linear regression of OTU counts on Th17:Treg ratios in those individuals with normoglycemia, prediabetes, and diabetes by SIgA coating. For those individuals with normoglycemia, p<0.958 and p<0.359 in the IgA$^+$ and IgA$^-$ compartments, respectively, for those individuals with prediabetes, p<0.819 and p<0.376 in the IgA$^+$ and IgA$^-$ compartments, respectively, and in those with diabetes, p<0.097 and p<0.567 in the IgA$^+$ and IgA$^-$ compartments, respectively.
(TIFF)

**S5 Fig. Principal coordinate analyses of UniFrac distances of the stool IgA-Biome by dichotomized Th17:Treg ratios across diabetes phenotype.** Principal coordinate analyses of unweighted UniFrac distances of the (**A, B, C**) SIgA$^+$ and (**C, D, E**) SIgA$^-$ compartments of

individuals dichotomized by their inflammatory profiles. (**A, D**) Normoglycemic and (**B, E**) prediabetic, and (**C, F**) diabetes. Statistical significance was determined using PERMANOVA. (TIFF)

**S6 Fig. Principal coordinate analyses of UniFrac distances of the salivary IgA-Biome by dichotomized Th17:Treg ratios across diabetes phenotype.** Principal coordinate analyses of unweighted UniFrac distances of the (**A, B, C**) SIgA$^+$ and (**C, D, E**) SIgA$^-$ compartments of individuals dichotomized by their inflammatory profiles. (**A, D**) Normoglycemic and (**B, E**) prediabetic, and (**C, F**) diabetes. Statistical significance was determined using PERMANOVA. (TIFF)

**S7 Fig. Comparisons of UniFrac distances between of the salivary IgA-Biome of normal and dysglycemic individuals categorized by Th17:Treg ratio.** Box plots compare unweighted UniFrac distances between all individuals within each inflammatory phenotype, respectively, by SIgA compartment and glycemic profile. $^*p<0.023$ determined using the Mann-Whitney U-test.
(TIFF)

## Acknowledgments

We would like to thank the participants for their willingness to be a part of our study and the amazing, tireless Starr County Health Studies' staff that are responsible for enrolling, consenting, and collecting the samples that allow us to conduct our various studies.

## Author Contributions

**Conceptualization:** Heather T. Essigmann, Eric L. Brown.

**Data curation:** Kristi L. Hoffman.

**Formal analysis:** Heather T. Essigmann, Craig L. Hanis, Eric L. Brown.

**Funding acquisition:** Craig L. Hanis, Eric L. Brown.

**Investigation:** Heather T. Essigmann, Goo Jun, David Aguilar, Craig L. Hanis, Herbert L. DuPont, Eric L. Brown.

**Methodology:** Heather T. Essigmann, Kristi L. Hoffman, Eric L. Brown.

**Project administration:** Joseph F. Petrosino, Craig L. Hanis, Eric L. Brown.

**Resources:** Kristi L. Hoffman, Goo Jun, Herbert L. DuPont, Eric L. Brown.

**Supervision:** Joseph F. Petrosino, Craig L. Hanis, Eric L. Brown.

**Validation:** Kristi L. Hoffman.

**Visualization:** Heather T. Essigmann, Joseph F. Petrosino, Craig L. Hanis, Eric L. Brown.

**Writing – original draft:** Heather T. Essigmann, Eric L. Brown.

**Writing – review & editing:** Heather T. Essigmann, Kristi L. Hoffman, Joseph F. Petrosino, Goo Jun, David Aguilar, Craig L. Hanis, Herbert L. DuPont, Eric L. Brown.

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
