## [Decision Letter · Decision Letter 0]

26 Aug 2021

PONE-D-21-17583

The impact of the Th17:Treg axis on the IgA-Biome across the glycemic spectrum

PLOS ONE

Dear Dr. Brown,

Thank you for submitting your manuscript to PLOS ONE. After careful consideration, we feel that it has merit but does not fully meet PLOS ONE’s publication criteria as it currently stands. Therefore, we invite you to submit a revised version of the manuscript that addresses the points raised during the review process.

We look forward to receiving your revised manuscript.

Kind regards,

Jane Foster, PhD

Academic Editor

PLOS ONE

1. Please ensure that your manuscript meets PLOS ONE's style requirements, including those for file naming. The PLOS ONE style templates can be found at https://journals.plos.org/plosone/s/file?id=wjVg/PLOSOne_formatting_sample_main_body.pdf and https://journals.plos.org/plosone/s/file?id=ba62/PLOSOne_formatting_sample_title_authors_affiliations.pdf.

“This work was funded by the National Institutes of Health grant R01DK116378 to ELB and CLH and by funds provided for by the Kelsey Research Foundation to HLD. We would like to thank the participants for their willingness to be a part of our study and the amazing, tireless Starr County Health Studies’ staff that are responsible for enrolling, consenting, and collecting the samples that allow us to conduct our various studies.”

“National Institutes of Health grant R01DK116378 to ELB and CLH.”

Additional Editor Comments (if provided):

Reviewers' comments:

Reviewer's Responses to Questions

**Comments to the Author**

1. Is the manuscript technically sound, and do the data support the conclusions?

Reviewer #1: Yes

Reviewer #2: Partly

2. Has the statistical analysis been performed appropriately and rigorously? 

Reviewer #1: Yes

Reviewer #2: Yes

3. Have the authors made all data underlying the findings in their manuscript fully available?

Reviewer #1: Yes

Reviewer #2: No

4. Is the manuscript presented in an intelligible fashion and written in standard English?

Reviewer #1: Yes

Reviewer #2: Yes

5. Review Comments to the Author

Reviewer #1: Overall this manuscript presents very exciting data that helps to clarify the intricate relationships between secretory signaling molecules, immune cells and the microbiomes in the context of glycemic control. However, I feel that perhaps the authors know their study and data so well, they are leaving out key details that help the reader to follow this complex system and multiple comparisons. I strongly encourage the authors to follow the comments below and also seek internal review from their own colleagues who similarly do not know the study as well to provide important feedback for additional clarity.

Abstract: I feel like the abstract is very heavy with background (though it does provide very helpful outlines of the field and give needed context to the manuscript). Can the authors work to try to remove small details in the Abstract to provide room for a little more explanation for the study at hand and the results? The bit about B cells could maybe be moved to the Introduction and given a little more space to explain the biological relationships, as well as saving some space in the Abstract.

Introduction: I fully appreciate that this is complex and difficult system to clearly explain. However, I think the Introduction could warrant a little more structure around explaining the different mucosal compartments/microbial environments under examination in the study: saliva vs. stool vs. blood (buffy coat). It may be difficult for some readers to follow how changes in one compartment vs. another or broad/systemic changes across all confers biological consequences. Give some time in the Intro to explain this and you save yourself headache in the later aspects of the paper (you also mention the consequences of differential change like this in the Discussion, so set it up better earlier).

Methods: What were the BMI differences between groups? Ideally, the participants could be matched on BMI as well but that is challenging. Please report the BMI for each sample group and determine any differences.

I would state about the plate runs and performance limits in only one location, either Method or Results (probably Method), no need to repeat the sentence in both places.

Results: Consider adding a Demographics reporting table that states the ranges for age, BMI, blood glucose measurements and %HbA1c values for each group. Are these exactly the same participants reported in Brown et al., 2020? I suggest rewording the Study Population description in that case to be more transparent that these are “pre-matched” from another study and not specifically matched for this current study. It seems like the current manuscript is a direct extension of this earlier manuscript and thus should perhaps be more clearly tied to the previous findings also. The mention of the “Presort” samples on Pg. 11, Line 213 definitely suggests that there is some overlap between samples/findings. This needs to be much more clear as to what samples/ranges/analyses are being directly compared in this current manuscript.

Figure and Figure Legend 7: I’m totally lost as to what is being presented when it’s telling me that I’m seeing “above median vs. above median” and “below median vs. below median” across these 3 sample sets…one of which contains the other two??? I’m very lost. Please add clarifying language, ideally to both the Figure itself and the legend.

Discussion: As much as I appreciate the reference to Hippocrates and the importance of the gut, the language might be a bit too strongly flowery for application in this manuscript. Strongly suggest toning that down.

Reviewer #2: The authors aimed to identify associations across the Th17:Treg-SIgA axis and glycemic profiles. The authors detected a relationship between Th17:Treg ratios and alpha diversity in the stool IgA-Biome of those with dysglycemia as well as changes in the composition of the stool and salivary microbiomes across glycemic profiles. This research is novel. The manuscript is well-written but requires some adjustments:

1) The abstract contains a lot of information on the background. However, the aim and the used methods are not mentioned clearly. The results are listed within a long sentence. Conclusion and outlook are missing. Please revise your abstract to increase readability and understanding.

2) The term IgA-Biome is interesting as it reflects specific bacteria as you have defined. Would it be perhaps an idea to adjust it to IgA-Bacteriome since I assume you are not including / assessing other microorganisms? Secondly, It would be useful to understand what bacteria are SIgA-coated/uncoated bacteria. The term has been not introduced when it appears first time in the introduction in line 72.

3) V4 represents a hypervariable region of 16S which could lead to taxonomic inaccuracy compared to sequencing the full 16S gene. Have you validated your results?

4) The figures were very pixeled and of low resolution. It was hard to evaluate them properly. Please update.

5) “Although not statistically significant, the opposite trend, i.e., decreasing richness with increasing Th17:Treg ratio, was observed in those with normoglycemia (Fig 2) indicating that bacteria richness is positively correlated with measures of inflammation only in those with abnormal blood sugar values.” It is unclear how the authors concluded here on a positive correlation.

6) It would be useful to quantify secretory IgA in this population and relate it to your findings, particularly the bacterial results, to understand the interrelationship better.

7) “8 with normal glycemia, 8 with prediabetes, and 8 with diabetes, were used in the current study”. It would be helpful for the reader to list the relevant metabolic parameters in this paper again. This can be added as a supplementary table.

8) Line 202 - relatively gaussian?

9) Fig. 1 - Th17:Treg ratios by diabetes profile: It would be still more accurate to differentiate between prediabetic and diabetic.

10) Fig. 2ff – it would be interesting to investigate whether there is a trajectory between normoglycemic and prediabetic towards diabetic. As such it would be helpful to re-graph all figures to include all 3 groups. From Fig. 2 onwards, it remains unclear why both groups prediabetes and diabetes are still combined.

11) “The observation that almost all human SIgA show evidence of somatic hypermutation and affinity maturation indicates that T-cell-dependent responses play a dominant role in shaping the IgA-Biome” The conclusion is unclear here. Please provide more context.

12) “Similar analysis of alpha diversity metrics in the salivary IgA-Biome did not identify significant associations” Please clarify what is meant with “similar analysis of alpha diversity”.

13) Please provide more clarification on what the Th17:Treg-SIgA axis entails. This could be done for instance by a graphic.

14) Line 374 - “categorized by T-cell phenotype”. Since not a range of T-cell phenotypes has been characterized, this should be revised.

15) “Faecalibacterium, a genus that only contains one species, F. prausnitzii, previously associated with chronic inflammation” That F. prausnitzii is not the only one species in this genera has been recently revised (de Filippis et al. 2020, Current Biology).

Typos

- Line 379: trended towards significance?

6. PLOS authors have the option to publish the peer review history of their article (what does this mean?). If published, this will include your full peer review and any attached files.

Reviewer #1: No

Reviewer #2: No

---

## [Author Response · Author response to Decision Letter 0]

17 Sep 2021

ONE-D-21-17583

The impact of the Th17:Treg axis on the IgA-Biome across the glycemic spectrum

PLOS ONE

1. Please ensure that your manuscript meets PLOS ONE's style requirements, including those for file naming. The PLOS ONE style templates can be found at https://journals.plos.org/plosone/s/file?id=wjVg/PLOSOne_formatting_sample_main_body.pdf and https://journals.plos.org/plosone/s/file?id=ba62/PLOSOne_formatting_sample_title_authors_affiliations.pdf.

Response:

Corrected.

“This work was funded by the National Institutes of Health grant R01DK116378 to ELB and CLH and by funds provided for by the Kelsey Research Foundation to HLD. We would like to thank the participants for their willingness to be a part of our study and the amazing, tireless Starr County Health Studies’ staff that are responsible for enrolling, consenting, and collecting the samples that allow us to conduct our various studies.”

“National Institutes of Health grant R01DK116378 to ELB and CLH.”

Response:

Funding-related text has been removed from the manuscript. The funding source number of R01DK116378 is correct. It will be corrected in the on-line portion of the submission.

Response:

All ‘data not shown’ phrases have been removed from the modified manuscript. Seven supplementary Figs and 2 new additional Figs have been added to the revised manuscript.

Response: 

The section in the Methods pertaining to the consent has been revised.

Comments to the Author

Reviewer #1: 

Overall this manuscript presents very exciting data that helps to clarify the intricate relationships between secretory signaling molecules, immune cells and the microbiomes in the context of glycemic control. However, I feel that perhaps the authors know their study and data so well, they are leaving out key details that help the reader to follow this complex system and multiple comparisons. I strongly encourage the authors to follow the comments below and also seek internal review from their own colleagues who similarly do not know the study as well to provide important feedback for additional clarity.

1. Abstract: I feel like the abstract is very heavy with background (though it does provide very helpful outlines of the field and give needed context to the manuscript). Can the authors work to try to remove small details in the Abstract to provide room for a little more explanation for the study at hand and the results? The bit about B cells could maybe be moved to the Introduction and given a little more space to explain the biological relationships, as well as saving some space in the Abstract.

Response:

The Abstract has been modified as recommended.

2. Introduction: I fully appreciate that this is complex and difficult system to clearly explain. However, I think the Introduction could warrant a little more structure around explaining the different mucosal compartments/microbial environments under examination in the study: saliva vs. stool vs. blood (buffy coat). It may be difficult for some readers to follow how changes in one compartment vs. another or broad/systemic changes across all confers biological consequences. Give some time in the Intro to explain this and you save yourself headache in the later aspects of the paper (you also mention the consequences of differential change like this in the Discussion, so set it up better earlier).

Response:

Thank you for the suggestion. A new paragraph in the introduction has been added to address this comment (Lines: 42-82). In addition, a new Fig 1 has been added to aid with the explanation of the Th17:Treg/SIgA Axis and the microbiome.

3. Methods: What were the BMI differences between groups? Ideally, the participants could be matched on BMI as well but that is challenging. Please report the BMI for each sample group and determine any differences.

Response:

The following section was added to the Materials (Lines 144-151):

‘Characterization of the gut and salivary IgA-Biomes and demographics of the study participants and were previously published by our group [13]. Briefly, 16.67% (4/24) of subjects were male and the mean age of participants was 47.75 years (standard deviation (SD) = 5.96). The body mass index (BMI) varied by diabetes phenotype (ANOVA = 0.05) with mean BMIs of 30.6 kg/m2 (SD = 4.8), 33.0 kg/m2 (SD = 3.0), and 37.7 kg/m2 (SD = 7.6) for those classified as normoglycemic, prediabetic, and diabetic, respectively. In the present study, the buffy coats were used to define the Th17:Treg ratios (see below) to establish correlations between the IgA-Biome and diabetes profiles.’

In addition, differences in BMI across groups were not statistically significant by Kruskal Wallis H-test (p=0.08); however, figures 2 A-B, figures 4 A-D, figures 6 A-B, and figures 7 A-D have been reanalyzed adjusting for BMI. Please see revised ‘Statistical analyses’ section in the Methods (Lines: 225-240).

4. I would state about the plate runs and performance limits in only one location, either Methods or Results (probably Methods), no need to repeat the sentence in both places.

Response:

This has been corrected and only appears in the Methods section as suggested.

5. Results: Consider adding a Demographics reporting table that states the ranges for age, BMI, blood glucose measurements and %HbA1c values for each group. Are these exactly the same participants reported in Brown et al., 2020? I suggest rewording the Study Population description in that case to be more transparent that these are “pre-matched” from another study and not specifically matched for this current study. It seems like the current manuscript is a direct extension of this earlier manuscript and thus should perhaps be more clearly tied to the previous findings also. The mention of the “Presort” samples on Pg. 11, Line 213 definitely suggests that there is some overlap between samples/findings. This needs to be much more clear as to what samples/ranges/analyses are being directly compared in this current manuscript.

Response:

We agree with the Reviewer that it is not clear that the IgA-Biome was established in our prior publication (Brown et al., 2020) [reference 13].

In the ‘Study population and sample collection’ in the Methods section the following sentence was added (Lines: 141-151)

‘Participants were age matched and their respective gut and salivary IgA-Biomes and demographics were previously published by our group [13]. Briefly, 16.67% (4/24) of subjects were male and the mean age of participants was 47.75 years (standard deviation (SD) = 5.96). The body mass index (BMI) varied by diabetes phenotype (ANOVA = 0.05) with mean BMIs of 30.6 kg/m2 (SD = 4.8), 33.0 kg/m2 (SD = 3.0), and 37.7 kg/m2 (SD = 7.6) for those classified as normoglycemic, prediabetic, and diabetic, respectively. In the present study, the buffy coats were used to define the Th17:Treg ratios (see below) to establish correlations between the IgA-Biome and diabetes profiles.’ 

In the ‘Epigenetic characterization of T-helper cell profiles’ section of the Methods the following sentence was added (Lines: 197-198):

‘Buffy coats were collected at the same time stool and saliva samples for IgA-Biome analyses were collected.’

In the ‘16S rRNA gene sequencing’ section of the Methods the following sentence was added (Lines: 182-184):

‘The IgA-Biome profiles defined for the 24 participants described in the present study were previously published by our group [13].’

We also agree that the description of the Presort is unclear. Prior to sorting samples into IgA+ and IgA- groups a portion of each sample was collected and sequenced with the sorted samples in parallel. This unsorted sample is referred to as the Presort. In the ‘16S RNA gene sequencing’ section of the Methods the following sentence was added (Lines: 178-180):

‘In addition to sequencing IgA+ and IgA- bacteria from each sample, unsorted (Presort) samples were collected for each sample prior to sorting and sequenced in parallel.”

6. Figure and Figure Legend 7: I’m totally lost as to what is being presented when it’s telling me that I’m seeing “above median vs. above median” and “below median vs. below median” across these 3 sample sets…one of which contains the other two??? I’m very lost. Please add clarifying language, ideally to both the Figure itself and the legend.

Response:

We agree with the Reviewer that this is a difficult analysis to explain. First, instead of using below the median or above the median we have now explained that above the median ratio is a ‘pro-inflammatory’ profile and below the median ratio is ‘anti-inflammatory. Second, the box plots to the Fig in question (now Fig 8) have been changed to ‘Pro-Inflammatory Phenotype’ or ‘Anti-Inflammatory Phenotype.’ The reason the data were presented as they were i.e., comparing ‘below vs. below’ to ‘above vs. above’ is because the UniFrac test measures the evolutionary distance (distances between communities as branch length percentage) between taxa in each ‘above’ or ‘below’ group. Therefore, it was written as, for example, ‘below vs. below’ because the test was determining distances between communities in the ‘below group’ and the same to those in the ‘above group’ by comparing distances within in each group, that is, above vs. above and below vs. below. That is all gone! In addition, a sentence was added to better explain UniFrac and the legend to the new Fig. 8 has been modified as well.

Lines: 396-401:

‘Comparisons of individuals grouped as having pro-inflammatory or anti-inflammatory profiles based on Th17:Treg ratios were used to investigate beta, or between sample, diversity of stool and salivary Presort samples using unweighted UniFrac distances, a distance metric that incorporates phylogenetic relatedness and multivariate statistical techniques to determine whether microbial communities are significantly different.’

New Fig 8 legend:

Fig 8. Comparisons of UniFrac distances between of the stool IgA-Biome of normal and dysglycemic individuals categorized by inflammatory phenotype. Box plots compare unweighted UniFrac distances between all individuals within each inflammatory phenotype, respectively, by SIgA compartment and glycemic profile. Normoglycemic subjects with a pro-inflammatory phenotype have SIgA+ bacterial community compositions more similar to each other than do those with an anti-inflammatory phenotype (*p<0.0152). Similarly, dysglycemic subjects with a pro-inflammatory phenotype have SIgA- bacterial community compositions more similar to each other than do those with an anti-inflammatory phenotype (**p<0.023). P-values determined using the Mann-Whitney U-test. 

7. Discussion: As much as I appreciate the reference to Hippocrates and the importance of the gut, the language might be a bit too strongly flowery for application in this manuscript. Strongly suggest toning that down.

Response: 

This sentence has been removed.

Reviewer #2: 

The authors aimed to identify associations across the Th17:Treg-SIgA axis and glycemic profiles. The authors detected a relationship between Th17:Treg ratios and alpha diversity in the stool IgA-Biome of those with dysglycemia as well as changes in the composition of the stool and salivary microbiomes across glycemic profiles. This research is novel. The manuscript is well-written but requires some adjustments:

1. The abstract contains a lot of information on the background. However, the aim and the used methods are not mentioned clearly. The results are listed within a long sentence. Conclusion and outlook are missing. Please revise your abstract to increase readability and understanding.

Response:

Reviewer 1 also had recommendations regarding the Abstract. The modified form attempts to encompass both Reviewers’ comments.

2. The term IgA-Biome is interesting as it reflects specific bacteria as you have defined. Would it be perhaps an idea to adjust it to IgA-Bacteriome since I assume you are not including / assessing other microorganisms? Secondly, it would be useful to understand what bacteria are SIgA-coated/uncoated bacteria. The term has been not introduced when it appears first time in the introduction in line 72.

Response:

We agree with the Reviewer that we are at this time only looking at bacteria. However, as new taxonomic sequences are added to the pipelines, additional taxa including fungi and viruses can be added to the analyses. We have also already published one manuscript using ‘IgA-Biome’ for consistency would like to maintain it as it is. The IgA-Biome is now part of the Abstract and of the revised Introduction.

3. V4 represents a hypervariable region of 16S which could lead to taxonomic inaccuracy compared to sequencing the full 16S gene. Have you validated your results?

Response:

While long read full length (FL) 16S sequencing was not available through our collaborators at the CMMR Sequencing Core at the time sequencing was performed, we fully agree that FL 16S information would provide improved taxonomic accuracy, including species/strain level assignments. Hence, future research with these and related samples from the larger cohort will use FL 16S sequencing or, pending greater availability of material, whole genome shotgun (WGS) sequencing, enabling validation of the results presented here. For the current work, we selected the V4 region to be consistent with previously published data on the IgA-Biome and the stool microbiome more generally, allowing us to make more broadly appropriate comparisons with the literature, and limited analysis to genus-level assessments as appropriate for short read amplicon data. 

4) The figures were very pixeled and of low resolution. It was hard to evaluate them properly.

Response:

All Figs have been uploaded with a minimum 1200 dpi.

5) “Although not statistically significant, the opposite trend, i.e., decreasing richness with increasing Th17:Treg ratio, was observed in those with normoglycemia (Fig 2) indicating that bacteria richness is positively correlated with measures of inflammation only in those with abnormal blood sugar values.” It is unclear how the authors concluded here on a positive correlation.

Response:

We think the confusion arises because the previous sentence is focused on those with dysglycemia and the sentence in question is supposed to be focusing on the normoglycemic group but ends with a mention of the association between Th17:Treg ratios and those with dysglycemia. The sentence now reads (Lines 280-282):

“Although not statistically significant, the opposite trend, i.e., decreasing richness with increasing Th17:Treg ratio, was observed in those with normoglycemia (Fig 3A; p<0.066).

6. It would be useful to quantify secretory IgA in this population and relate it to your findings, particularly the bacterial results, to understand the interrelationship better.

Response:

We agree with the Reviewer; however, we feel this is beyond the scope of the present study. In addition, we feel that the focus of the IgA work is what is to examine taxa that are SIgA coated or uncoated. We are not sure that knowing the amount of SIgA would provide additional information. The percent coating of bacteria for each participant is reported in our earlier publication [ref 13].

7. “8 with normal glycemia, 8 with prediabetes, and 8 with diabetes, were used in the current study”. It would be helpful for the reader to list the relevant metabolic parameters in this paper again. This can be added as a supplementary table.

Response:

The requested information has now been added (Lines: 141-151):

‘Stool, saliva, and buffy coat samples collected upon enrollment from an age matched subset of 24 participants, 8 with normal glycemia, 8 with prediabetes, and 8 with diabetes [13]. Characterization of the gut and salivary IgA-Biomes and demographics of the study participants and were previously published by our group [13]. Briefly, 16.67% (4/24) of subjects were male and the mean age of participants was 47.75 years (standard deviation (SD) = 5.96). The body mass index (BMI) varied by diabetes phenotype (ANOVA = 0.05) with mean BMIs of 30.6 kg/m2 (SD = 4.8), 33.0 kg/m2 (SD = 3.0), and 37.7 kg/m2 (SD = 7.6) for those classified as normoglycemic, prediabetic, and diabetic, respectively. In the present study, the buffy coats were used to define the Th17:Treg ratios (see below) to establish correlations between the IgA-Biome and diabetes profiles.’

8. Line 202 - relatively gaussian?

Response:

‘Relatively’ has been removed. The sentence now reads (Lines: 263-264):

‘Th17:Treg T-cell ratios were normally distributed (Shapiro-Wil test of normality p-value = 0.08) with a median of 1.79 and mean of 1.90 (SD: 057, range: 0.82 – 3.24).’

9. Fig. 1 - Th17:Treg ratios by diabetes profile: It would be still more accurate to differentiate between prediabetic and diabetic.

Response:

We now show these data in Figure 2. Fig 2A is the same as the original Fig 1 and Fig 2B shows the ratios for those with prediabetes and diabetes.

10. Fig. 2ff – it would be interesting to investigate whether there is a trajectory between normoglycemic and prediabetic towards diabetic. As such it would be helpful to re-graph all figures to include all 3 groups. From Fig. 2 onwards, it remains unclear why both groups prediabetes and diabetes are still combined.

Response:

The prediabetic and diabetics had similar associations with the Th17:Treg ratios and were therefore combined into the dysglycemia group to increase sample sizes and our ability to detect differences between these phenotypes; however, we agree with the Reviewer and now include either in the manuscript or in the supplementary Figs data for the prediabetes and diabetes groups.

11. “The observation that almost all human SIgA show evidence of somatic hypermutation and affinity maturation indicates that T-cell-dependent responses play a dominant role in shaping the IgA-Biome” The conclusion is unclear here. Please provide more context.

Response:

We have added an additional sentence below to hopefully clarify this statement (Lines: 340-346):

‘More specifically as it relates to IgA production, B-cells of the mucosal lymphoid tissues are different form B-cells populating non-mucosal lymphoid tissues in that they can undergo class switching from IgM/IgD to IgA without T-cell help; however, like B-cells at all other anatomic sites, mucosal B-cells cannot produce high affinity antibodies, the products of germinal center reactions and somatic hypermutation, without the aid of T-cells [11, 74].’

12. “Similar analysis of alpha diversity metrics in the salivary IgA-Biome did not identify significant associations” Please clarify what is meant with “similar analysis of alpha diversity”.

Response:

‘Similar’ was removed from the sentence and now reads as follows (Lines 356-358):

‘Analysis of alpha diversity metrics in the salivary IgA-Biome did not identify significant associations (Fig S4A and S4B).’

13. Please provide more clarification on what the Th17:Treg-SIgA axis entails. This could be done for instance by a graphic.

Response:

We have generated a new Fig 1 graphic describing the Th17:Treg-SIgA axis.

14. Line 374 - “categorized by T-cell phenotype”. Since not a range of T-cell phenotypes has been characterized, this should be revised.

Response:

This terminology has been modified throughout the manuscript. The sentence now reads as follows (Lines: 475-476):

‘We also observed significant differences in community composition (beta diversity) between individuals categorized by their inflammatory profiles in the stool…’

15. “Faecalibacterium, a genus that only contains one species, F. prausnitzii, previously associated with chronic inflammation” That F. prausnitzii is not the only one species in this genera has been recently revised (de Filippis et al. 2020, Current Biology).

Response:

Many thanks for catching that! The Filippis reference has been added and this portion of the Discussion has been modified as follows (Lines: 498-502):

‘The only genus present in both SIgA+ and SIgA- compartments was Faecalibacterium, a genus that only contains 22 species, 12 of which inhabit the human gut [103]. Although F. prausnitzii has been associated with chronic inflammation, other Faecalibacterium species are leading candidates for the development of probiotics and biotherapeutics [103, 104].

Typos

- Line 379: trended towards significance?

Response:

This sentence now reads as follows (Lines 480-482):

A similar but not significant trend in the SIgA+ fraction of those with normal blood sugar values may indicate that changes to SIgA-targeted communities is heightened in the context of an inflammatory disease like type 2 diabetes.

---

## [Decision Letter · Decision Letter 1]

6 Oct 2021

The impact of the Th17:Treg axis on the IgA-Biome across the glycemic spectrum

PONE-D-21-17583R1

Dear Dr. Brown,

We’re pleased to inform you that your manuscript has been judged scientifically suitable for publication and will be formally accepted for publication once it meets all outstanding technical requirements.

Kind regards,

Jane Foster, PhD

Academic Editor

PLOS ONE

Additional Editor Comments (optional):

Reviewers' comments:

Reviewer's Responses to Questions

**Comments to the Author**

1. If the authors have adequately addressed your comments raised in a previous round of review and you feel that this manuscript is now acceptable for publication, you may indicate that here to bypass the “Comments to the Author” section, enter your conflict of interest statement in the “Confidential to Editor” section, and submit your "Accept" recommendation.

Reviewer #1: All comments have been addressed

Reviewer #2: All comments have been addressed

2. Is the manuscript technically sound, and do the data support the conclusions?

Reviewer #1: Yes

Reviewer #2: Yes

3. Has the statistical analysis been performed appropriately and rigorously? 

Reviewer #1: Yes

Reviewer #2: Yes

4. Have the authors made all data underlying the findings in their manuscript fully available?

Reviewer #1: Yes

Reviewer #2: Yes

5. Is the manuscript presented in an intelligible fashion and written in standard English?

Reviewer #1: Yes

Reviewer #2: Yes

6. Review Comments to the Author

Reviewer #1: My compliments to the authors, this is an excellent revision. The manuscript reads much more clearly. Great work here.

Reviewer #2: Dear authors,

Thank you for your comprehensive response to the raised concerns. I have no further questions.

7. PLOS authors have the option to publish the peer review history of their article (what does this mean?). If published, this will include your full peer review and any attached files.

Reviewer #1: **Yes: **Brittany L. Mason

Reviewer #2: No

---

## [Editor Report · Acceptance letter]

12 Oct 2021

PONE-D-21-17583R1 

The impact of the Th17:Treg axis on the IgA-Biome across the glycemic spectrum 

Dear Dr. Brown:

I'm pleased to inform you that your manuscript has been deemed suitable for publication in PLOS ONE. Congratulations! Your manuscript is now with our production department. 

Kind regards, 

on behalf of

Dr. Jane Foster 

Academic Editor

PLOS ONE